# The *Streptomyces viridochromogenes* product template domain represents an evolutionary intermediate between dehydratase and aldol cyclase of type I polyketide synthases

Yuanyuan Feng[1,3], Xu Yang[1,3], Huining Ji[1], Zixin Deng [1], Shuangjun Lin[1] & Jianting Zheng [1,2✉]

The product template (PT) domains act as an aldol cyclase to control the regiospecific aldol cyclization of the extremely reactive poly-β-ketone intermediate assembled by an iterative type I polyketide synthases (PKSs). Up to now, only the structure of fungal PksA PT that mediates the first-ring cyclization via C4–C9 aldol cyclization is available. We describe here the structural and computational characterization of a bacteria PT domain that controls C2–C7 cyclization in orsellinic acid (OSA) synthesis. Mutating the catalytic H949 of the PT abolishes production of OSA and results in a tetraacetic acid lactone (TTL) generated by spontaneous O-C cyclization of the acyl carrier protein (ACP)-bound tetraketide inter-mediate. Crystal structure of the bacterial PT domain closely resembles dehydrase (DH) domains of modular type I PKSs in the overall fold, dimerization interface and His-Asp catalytic dyad organization, but is significantly different from PTs of fungal iterative type I PKSs. QM/MM calculation suggests that the catalytic H949 abstracts a proton from C2 and transfers it to C7 carbonyl to mediate the cyclization reaction. According to structural similarity to DHs and functional similarity to fungal PTs, we propose that the bacterial PT represents an evolutionary intermediate between the two tailoring domains of type I PKSs.

[1] State Key Laboratory of Microbial Metabolism, School of Life Sciences and Biotechnology, Shanghai Jiao Tong University, Shanghai, China. [2] Joint International Research Laboratory of Metabolic & Developmental Sciences, Shanghai Jiao Tong University, Shanghai, China. [3] These authors contributed equally: Yuanyuan Feng, Xu Yang. ✉email: jtzheng@sjtu.edu.cn

Polyketides are structurally diverse natural products, constituting one of the most important sources of bioderived pharmaceuticals, for example, antibiotics, antitumor agents, and immunosuppressants[1–3]. They are biosynthesized from coenzyme A (CoA)-activated carboxylic acids by multifunctional megaenzymes or multisubunit enzyme complexes called polyketide synthases (PKSs) that resemble fatty acid synthases (FASs), using a β-ketoacyl synthase (KS), acyl transferase (AT) and acyl carrier protein (ACP). The resulting ACP-bound poly-β-keto thioester intermediates are ultimately converted to highly diversified compounds by optional tailoring domains. PKSs are grouped into different types according to their biosynthetic machinery architectures, including modular type I PKSs, iterative type I PKSs, iterative type II PKS, and the ACP-independent freestanding iterative type III PKSs[4]. Modular type I PKSs are widely distributed in bacteria, comprising multiple sets of domains that are organized into modules[3]. In contrast, the iterative type I PKSs are usually found in fungal species, repeatedly reusing one module that is highly programmed to generate polyketides with diverse structural features by using different domain combinations in each iteration[5].

Iterative type I PKSs are further classified as non-reducing (NR), partial-reducing (PR), and highly-reducing (HR) PKSs according to the function and phylogeny[6–8]. NR-PKSs synthesize aromatic polyketides that have important biological activities, exemplified by the environmental carcinogen aflatoxin B1. The poly-β-ketone backbone assembled by a NR-PKS is extremely reactive. Spontaneous cyclization must be suppressed to form the unique cyclization pattern observed in the final product. PksA involved in biosynthesis of aflatoxin B1 is a model system to investigate NR-PKSs. Functional dissection of PksA reveals a unique product template (PT) domain between AT and ACP domains, which functions as an aldol cyclase mediating the regioselective cyclization of a hexanoyl-primed octaketide intermediate via C4–C9/C2–C11 aldol condensations to generate a key intermediate of aflatoxin biosynthesis, norsolorinic acid (Fig. 1)[9,10].

NR-PKSs are divided into three basal subclades (subclades I–III) according to a phylogenetic analysis of KS sequences[11]. PTs embedded in subclades I and II are classified into five groups based on a phylogenetic analysis of those associated with known aromatic polyketides[12]. The well-studied PksA PT and other NR-PKS PTs involved in C4–C9/C2–C11 cyclization belong to Group IV. PTs of group I perform the regioselective C2–C7 cyclization of a tetraketide primed by different starter units and are embedded in NR-PKSs that synthesize polyketides containing a single aromatic ring, exemplified by the orsellinic acid synthase (OSAS) from *Aspergillus nidulans*[13,14]. PTs of group II catalyze

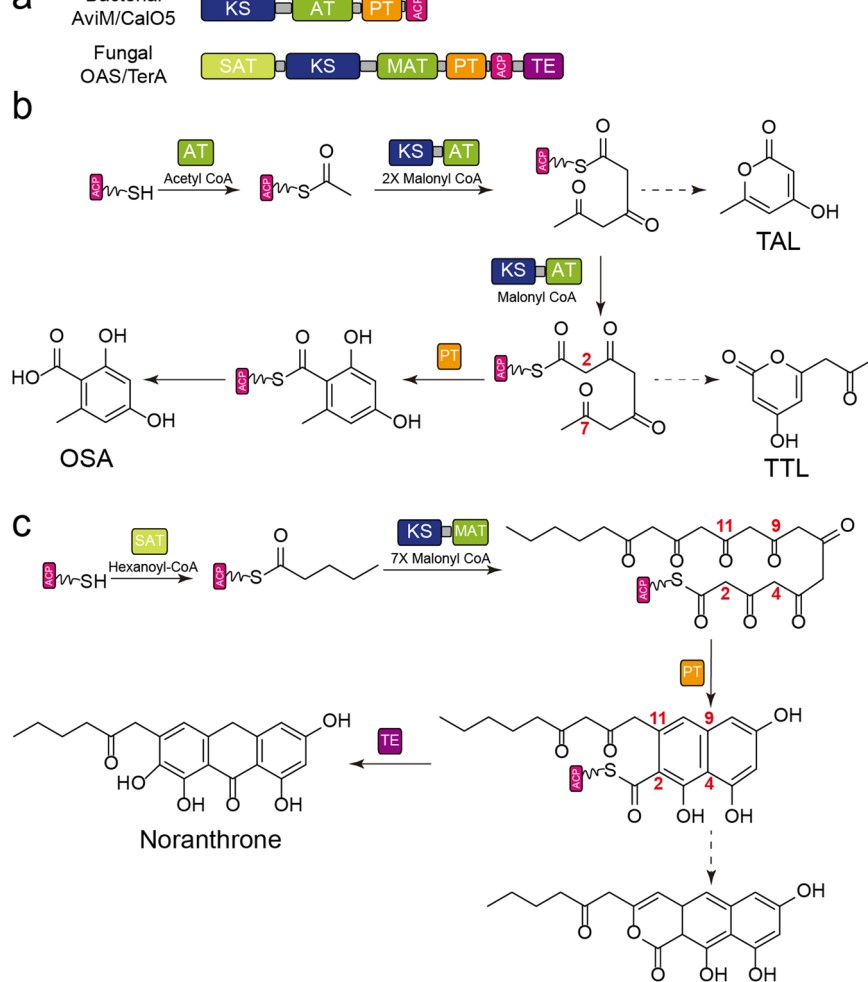

**Fig. 1 Biosynthesis reaction of OSA and Aflatoxin. a** The domain compositions of bacterial and fungal OSASs. Bacterial OSASs lack the starter-unit acyl transferase (SAT) and the thioesterase (TE) domains. **b** Biosynthesis of OSA in bacteria. Triacetic acid lactone (TAL) released after two condensation steps and tetraacetic acid lactone (TTL) are generated in the absence of PT due to the spontaneous O–C cyclization. AviM PT catalyzes the C2–C7 cyclization. **c** Biosynthesis reaction of Aflatoxin. PksA PT catalyzes the C4–C9 and C2–C11 cyclization.

the C2–C7 cyclization of pentaketide intermediates to form the first aromatic ring of tetrahydroxynaphthalene (THN) compounds, of which the second ring is formed by thioesterase-catalyzed Claisen cyclization[15]. PTs of group III and group V mediate cyclization of longer intermediates to form polyketides containing multiply fused-ring structures. Group III PTs have C2–C7 regioselectivity while group V PTs cyclize the nascent polyketide via C6–C11/C4–C13 regioselectivity[12]. NR-PKSs in subclade III contain an additional methyltransferase (CMeT) domain compared to those in subclade I and II and produce aromatics with C2–C7 cyclization regioselectivity. The PTs of subclade III NR-PKSs are classified into groups VI and VII[16]. Recently, phylogenetic analysis of all known fungal NR-PKSs reveals a new group VIII PT domains[17].

Only the structure of PksA PT has been reported, displaying a distinct double hotdog (DHD) fold, a variant of that observed in the dehydrase (DH) domains of modular type I PKSs[10,18]. PksA PT is dimeric in crystal structures, indicating the role of PT domains in dimerization of fungal NR-PKSs. A PT-specific sequence insertion is involved in dimerization of the two monomers and conserved in fungal NR-PKSs (Supplementary Fig. 1). Each monomer contains an internal pocket that extends 30 Å from the surface to the bottom for substrate binding and is visually divided into the phosphopantetheine (PPANT)-binding region, the cyclization chamber, and the hydrophobic hexyl-binding region. As the only PT domain that has been structurally characterized, PksA PT provides a vital model for understanding the high regioselective aldol cyclization of poly-β-ketone intermediates assembled by NR-PKSs. The PksA PT active site is a His-Asp catalytic dyad located at the cyclization chamber. In a binary structure of PksA PT complexed with a bisisoxazole model of the experimentally inaccessible poly-β-ketone intermediate (PDB code 5KBZ), the D1543 polarizes the H1345 by hydrogen bond to Nε so that the Nδ of H1345 functions as a basic nitrogen to deprotonate the substrate in the proposed mechanism of PT-catalyzed cyclization (Supplementary Fig. 2)[18].

The proposed DH mechanism based on a crystal structure of PpsC DH (PDB code 5NJI) in complex with an α,β-double bond substrate in a *trans* configuration also involves a His-Asp catalytic dyad, including an α-deprotonation and a β-hydroxyl elimination stages[19]. In the PpsC DH-catalyzed dehydration, the Nδ of catalytic H959 forms a hydrogen bond with the backbone carbonyl of V996 so that the Nε functions as the basic nitrogen to deprotonate the substrate. The D1129 side chain acts as a general acid by interacting with the β-hydroxyl group of the substrate (Supplementary Fig. 2b). $^{15}$N NMR assays of *E. coli* FabA, a β-hydroxydecanoyl thioester DH, demonstrate that the catalytic H70 is held in the Nδ-H tautomeric form by hydrogen bonding, whereas the Nε is used for proton subtraction[20]. In a crystal structure of FabA-3-decynoyl-N-acetylcysteamine complex, the Nε of the catalytic H70 is covalently modified by the inactivator. QM/MM studies of mammalian FAS DH (2VZ9) also supports that the Nε of catalytic H887 abstract a proton from C2 in the dehydration reaction.

Orsellinic acid (OSA) is synthesized in bacteria and, in most case, functions as a building block that is further incorporated into biosynthesis of more complex natural products, for example, avilamycin produced by *Streptomyces viridochromogenes*[21] and calicheamicin produced by *Micromonospora echinospora*[22]. Cloning and analysis of biosynthetic gene clusters suggests that AviM and CalO5 are responsible for the assembly of OSA. Cultures of *Streptomyces lividans* TK24 and *Streptomyces coelicolor* CH999 containing AviM can produce OSA, confirming the function of AviM as OSAS[21]. Compared to fungal OSASs[14], AviM and CalO5 lack the starter-unit acyl transferase (SAT) domain loading a primer for polyketide extension and the

thioesterase (TE) domain releasing OSA from the PPANT group of the ACP domain (Fig. 1a). Up to now, no bacterial PT has been characterized. Here, we confirm that AviM PT domain functions as an aldol cyclase to catalyze C2–C7 cyclization in OSA synthesis by comparing the products generated by AviM and its mutants. A tetraacetic acid lactone (TTL) is generated by spontaneous O–C cyclization of the ACP-bound tetraketide backbone when the catalytic H949 of the PT domain is replaced by Ala or Phe. The structure of AviM PT, solved by selenomethionine incorporation and single-wavelength anomalous dispersion (SAD) phasing, closely resembles dehydratase (DHs) of modular type I PKSs in the overall fold, dimerization interface and catalytic His-Asp dyad organization. In the AviM PT structure, the Nδ of H949 is involved in a hydrogen bond with the backbone carbonyl oxygen of I956, indicating that the catalytic H949 likely uses its Nε as the basic nitrogen to deprotonate the substrate. QM/MM calculation demonstrates that the catalytic H949 of AviM PT abstracts a proton from C2 and transfers it to C7 carbonyl in the aldol cyclization reaction. In a phylogenetic tree comprising fungal PTs, bacterial PTs and DHs of modular type I PKS, fungal PTs forms the first clade while the second clade splits into a subclade containing DHs and a subclade comprising bacterial PTs. The bacterial PT subclade falls between the DH subclade and the fungal PT clade. The structural similarity to DHs, the functional similarity to fungal PTs and the phylogenetic analysis suggesting that bacterial PTs may represent an evolutionary intermediate between DH and fungal PT domains.

## Results

**AviM PT catalyzes C2–C7 cyclization of OSA.** AviM is previously shown to synthesize OSA in *S. lividans* TK24 and *S. coelicolor* CH999[21]. Here we used *E. coli* K207-3[23], an *E. coli* B-derived strain developed for the heterologous expression of polyketide biosynthetic genes, to investigate the function of AviM in vivo. The DNA fragments encoding AviM was amplified by PCR and ligated into pET28a. The resulting pET28a-*aviM* was transformed into *E. coli* K207-3 to give the recombinant strain for OSA production, while the *E. coli* K207-3 carrying the vector pET28a was used as the control strain. LC–MS analysis revealed that cultures of *E. coli* K207-3 containing pET28a-*aviM* were able to produce OSA (Fig. 2).

The catalytic dyad of PksA PT is composed of His1345 and Asp1543[18]. The H949 residue of AviM PT, corresponding the catalytic H1345 of PksA PT, was replaced by site-specific mutagenesis with Ala or Phe. LC–MS analysis showed that the mutations of H949A and H949F abolished OSA production of AviM in *E. coli* K207-3, confirming the importance of H949 in catalysis. However, the extracted ion chromatography (EIC) revealed another product that had the same molecular weight but different retention time relative to OSA. MetFrag was used to identified this product with a relative mass deviation of 5 ppm and an absolute mass deviation of 0.002 Da as thresholds[24]. Tetraacetic acid lactone (TTL) was retrieved from PubChem as a potential candidate (Supplementary Fig. 3). Subsequently, the product was identified as TTL by comparing with the standard prepared by organic synthesis (Fig. 2a). Mutating D1543 of PksA PT to alanine results in no detectable activity[10], whereas the corresponding D1104A mutant of AviM PT retains most of the activity. A H949A/D1104A double mutant was generated and showed product profile similar to that of H949A mutant. These results suggests that D1104 of AviM may participate in catalysis indirectly. Obviously, the functions of the KS, AT, and ACP domains are not affected by mutations in the PT domain. The TTL product is generated by spontaneous O–C cyclization of the ACP-bound tetraketide backbone in the absence of PT domain

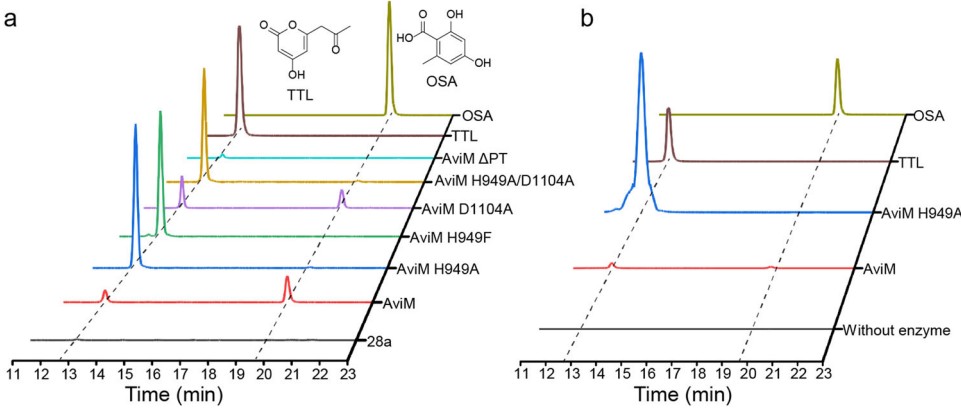

**Fig. 2 In vivo and in vitro assays of AviM-catalyzed OSA biosynthesis. a** *E. coli* K207-3 expressing AviM synthesizes OSA. Mutating the catalytic H949 abolishes OSA production, but results in TTL from spontaneous O-C cyclization of the ACP-bound tetraketide backbone. Mutating D1104 slightly increases TTL production. The product profile of H949A/D1104A double mutant is similar to that of H949A mutant. Deletion of PT domain from AviM abolishes OSA production whereas a small amount of TTL is still generated. **b** OSA is produced when AviM is incubated with acetyl-CoA and malonyl-CoA in vitro, while TTL is produced by AviM mutant. The traces show the extracted ion counts (m/z: 167.0350) monitored in the negative mode.

(Fig. 1b). In contrast, a C-C bond is formed between C2 and C7 by aldol cyclization in the presence of PT domain, followed by aromatization, to yield final OSA product. Analysis of the LC–MS data of the wild-type AviM also revealed the presence of TTL, whereas the production of TTL was significantly increased in H949A, H949F, and H949A/D1104A mutants (Fig. 2a). Interestingly, a triacetic acid lactone (TAL, **2**) released after two condensation steps was also observed in the cultures of the wild-type AviM and mutating the catalytic H949 also increased the production of the TAL (Supplementary Fig. 4). We also removed the PT domain from AviM. The resulting AviM ΔPT mutant lost the ability to produce OSA but still produced a small amount of TTL.

In vitro functional assays of AviM were carried out to exclude the possible effects of other enzymes encoded in *E. coli* K207-3 genome. The recombinant AviM was expressed in *E. coli* BAP1 containing a *sfp* gene from *Bacillus subtilis*[25]. The purified AviM (Supplementary Fig. 5) was further incubated with coenzyme A (CoA) and sfp enzyme to ensure the phosphopantetheinylation of all proteins. Trace amount of OSA was indeed produced when AviM was incubated with acetyl-CoA and malonyl-CoA in vitro (Fig. 2b). These in vitro and in vivo results confirm that AviM alone can mediate the assembly and release of OSA. Replacing the putative catalytic H949 by the Ala residue destroyed the aldol cyclase activity of AviM PT domain. TTL instead of OSA was produced by the AviM H949A mutant. In short, these in vivo and in vitro assays establish that the PT domain of AviM functions as an aldol cyclase to control the C2–C7 cyclization in OSA biosynthesis. Consequently, we proposed a OSA biosynthesis pathway presented in (Fig. 1b). The poly-β-keto chain elongation first starts with an acetyl-CoA building block. After two iterative rounds of decarboxylative Claisen condensations using malonyl building blocks, some intermediates are released by spontaneous O–C cyclization to yield a shunt product TAL. The remaining intermediates are condensed with the third malonyl-CoA molecule to form the ACP-bound tetraketide intermediate which were cyclized between C2 and C7 by the PT domain and released as OSA.

**Overall structure of AviM PT resembles DHs instead of PksA PT**. PT domains catalyze the first-ring cyclization via three common patterns (C2–C7, C4–C9, and C6–C11). Up to now, only PksA PT (group IV, PDB code 5KBZ) that cyclize the first ring via C4–C9 regiospecificity has been structurally

| Table 1 Data collection and refinement statistics. | | |
| --- | --- | --- |
| **Data collection** | | |
| Dataset | Native AviM PT | Se-Met Protein |
| Wavelength (Å) | 0.97853 | 0.97853 |
| Space group | *P*21 | *P*21 |
| Cell dimensions | | |
| a,b,c (Å) | 51.679, 58.708, 82.502 | 51.433, 58.556, 81.741 |
| α, β, γ (°) | 90.000, 96.275, 90.000 | 90.000, 95.907, 90.000 |
| Resolution (Å) | 50-2.0 (2.00-2.03) | 50-2.70 (2.75-2.70) |
| $R_{merge}$ | 0.062 (0.445) | 0.129 (0.580) |
| $I/\sigma I$ | 13.7 (2.42) | 11.1 (2.0) |
| Completeness (%) | 98.3 (97.8) | 99.9 (99.6) |
| Redundancy | 3.4 (3.5) | 3.4 (3.0) |
| **Phasing** | | |
| $R_{anom}/R_{p.i.m}$[1] | | 2.00 |
| d''/Sig[2] | | 0.66 |
| **Refinement** | | |
| Resolution (Å) | 50-2.0 | |
| No. reflections | 31071 | |
| $R_{work}/R_{free}$ | 0.2022/0.2528 | |
| No. atoms | 3744 | |
| B-factors (Å) | 49.743 | |
| R.m.s. deviations | | |
| Bond lengths (Å) | 0.01 | |
| Bond angles (°) | 0.783 | |

[1], [2] Calculated by program SHELXD in the CCP4 in the 50.00–2.70 Å resolution range.

characterized, hindering attempts to rationally control the cyclization patterns of poly-β-keto backbones. Our in vivo and in vitro functional assays revealed that AviM PT controls the C2–C7 cyclization in OSA biosynthesis. To understand the structural basis of the aldol cyclization steps, we solved the crystal structure of the AviM PT domain.

Diffraction-quality crystals were obtained by sitting-drop method using polyethylene glycol 4000 as a precipitant. The protein crystallized in space group *P*2₁ with two independent monomers per asymmetric unit. The structure was solved to 2.0-Å resolution through selenomethionine incorporation and SAD phasing (Table 1). The histidine-tag and the first ten residues are invisible in both monomers. Residues 946-950 and 1055-1069 are invisible in monomer B but visible in monomer A. The two monomers in the asymmetric unit have essentially identical folds

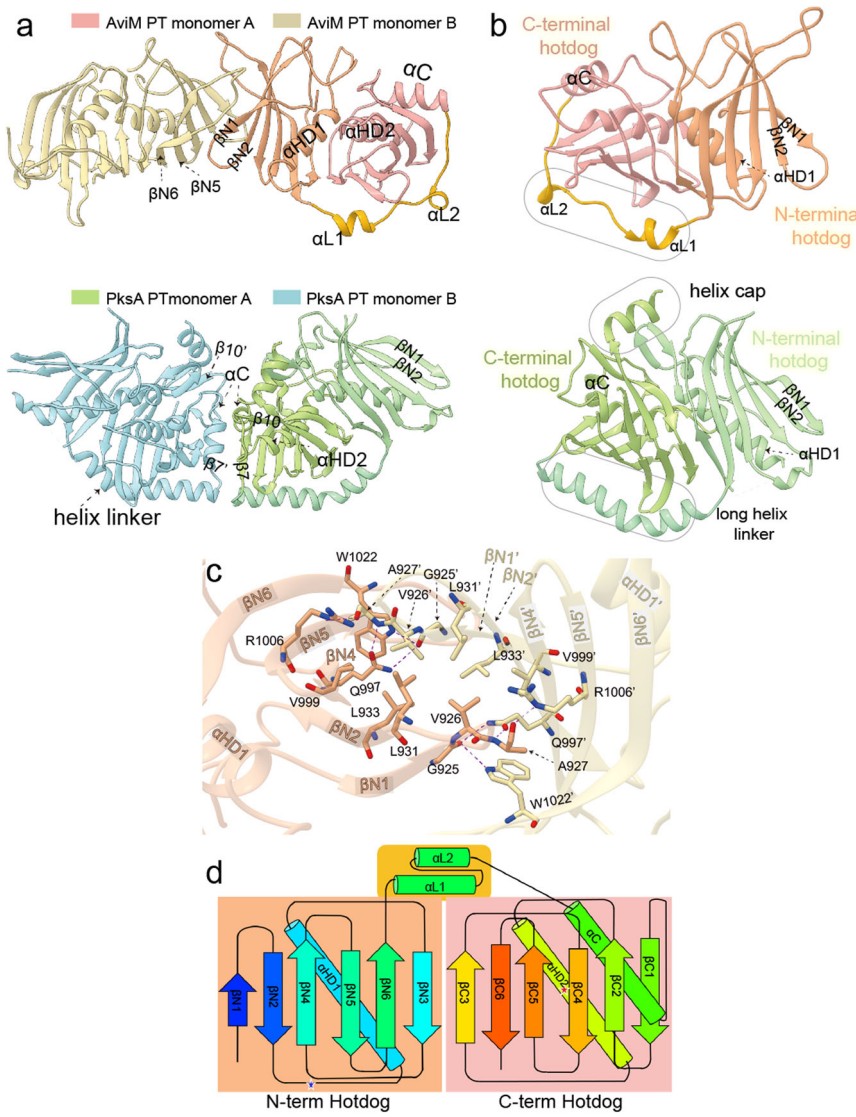

**Fig. 3 Overall structure of of AviM PT. a** The dimerization interface of AviM PT is different from that of PksA PT (PDB: 5KBZ). The two dimerization interfaces are basically 180° opposite. AviM PT dimerize via N-terminal hotdog while PksA PT dimerizes via C-terminal hotdog. **b** Differences between AviM PT monomer and PksA PT monomer. Compared to AviM PT monomer, PksA PT monomer contains a bent helix linker, a longer αHD1 helix, longer β1 and β2 strands, and a C-terminal helix capping substrate binding tunnel. **c** Residues located on dimer interface of AviM PT. Hydrogen bonds contributing to dimerization are shown as purple dashed lines. **d** The topology structure of AviM PT. The monomer has a N-terminal hotdog, a C-terminal hotdog and a linker containing two short α helices.

with a rmsd of 0.4 Å for Cα positions (Supplementary Fig. 6). An automatic search of AviM PT for structurally homologous proteins using the Dali server[26] revealed low sequence identity (less than 20%) to all previously reported structures. Interestingly, AviM PT shows more structural similarity to dehydratase (DH) domains than to PksA PT. Each monomer comprises a DHD fold of DH domains (Fig. 3), characterized by a continuous antiparallel β-sheet curving around two α-helices (αHD1 from the N-terminal and αHD2 from the C-terminal hotdog folds). The C-terminal hotdog (residues 1056-1174) possesses an additional helix (αC) comparted to the N-terminal hotdog (residues 915-1032). These two hotdogs are connected by a 23-residue-long flexible linker containing two short helices (αL1 and αL2). AviM PT most closely resembles the DH domain from CurF module of curacin PKS (PDB code 3KG6, 2.4 Å Cα rmsd; 19% sequence identity)[27]. The most notable difference is in the C-terminal hotdog of CurF DH which contains an additional $3_{10}$ helix and the connector between N-terminal and C-terminal

hotdog which lacks a linker helix (Supplementary Fig. 7). The PksA PT is ~70-residue larger than the AviM PT. They are superimposed with a rmsd of 3.2 Å for 147 (of 260 possible) Cα positions. Compared to AviM PT, four obvious differences are observed in PksA PT (Fig. 3b): (1) the N-terminal hotdog and the C-terminal hotdog are connected by a bent long helix; (2) the αHD1 from the N-terminal hotdog is two turns longer; (3) the βN1 and βN2 strands of the N-terminal hotdog are much longer; and (4) the ten C-terminal residues of the PksA PT form a helix that cap the substrate binding tunnel.

The two monomers in the asymmetric unit dimerize through N-terminal β strands and bury a surface area of 665 Å² (calculated by PDBe PISA). The relatively large dimer interface and the observed migration behavior as a dimer on the size exclusion column suggest that the PT domain is likely involved in oligomerization of AviM (Supplementary Fig. 8). The β-hairpin turns between βN1 and βN2, and between βN5 and βN6 contribute significantly to the dimer interface (Fig. 3a). Residues

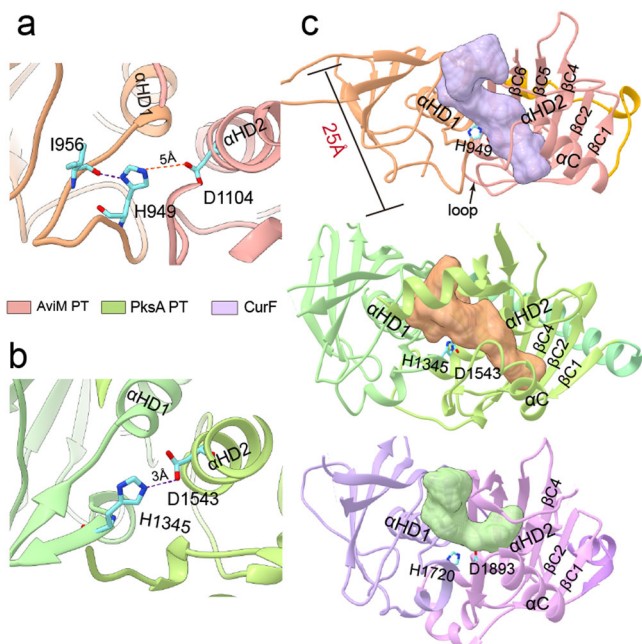

**Fig. 4 The active site of AviM PT. a** The Nδ of the catalytic H949 forms a hydrogen bond with the backbone carbonyl of I956 in AviM PT. **b** The Nε of the catalytic H1345 forms a hydrogen bond with the side chain of the catalytic D1543 in PksA PT. **c** The substrate tunnel of AviM PT is different from that of PksA PT and CurF. The cartoon structure of AviM PT, PksA PT and CurF are shown in orange, green and purple, respectively. The substrate tunnel of AviM PT, PksA PT and CurF are shown in purple, orange and green, respectively.

from the two monomers AviM PT that are within 4.5 Å from each other at the binding interface were recognized as interfacial residues. The dimer interface of AviM PT is dominated by hydrophobic interactions and hydrogen bonds (Fig. 3c). The residues of V926, L931, L933 and V999 create a hydrophobic core between the βN1, βN2 and βN4 sheets of the two monomers. The distance between the donor and acceptor atoms of a hydrogen bond was defined as less than 3.5 Å. According to this criterion, several hydrogen bonds are observed between the two monomers. The side chain of Q997 forms hydrogen bonds with the backbone NH of A927, and the backbone carbonyl oxygen of G925. The backbone carbonyl oxygen of G925 also makes a hydrogen bond with the side chain of W1022, whereas the backbone carbonyl oxygen of A927 forms hydrogen bonds with the side chain of R1006. Deleting the PT domain of AviM (residues 915-1174) abolished the production of both OSA and TTL (Fig. 2a), suggesting that the deletion of PT probably caused an architectural change rendering the megaenzyme inactive. The angle between the two monomers is ~180° in CurF DH and this extended conformation has been observed in all previously reported structures of PKS DH domains. However, the two monomers of AviM PT are obviously in a different relative orientation, forming an angle of ~150° (Supplementary Fig. 7b). Consistent with AviM PT, β sheets and β-hairpins of N-terminal hotdog of CurF DH participated in dimerization via hydrophobic interactions and hydrogen bonds (Supplementary Fig. 7c). The residues I1694, L1696, I1699 of both monomers form a hydrophobic core at the dimer interface. The side chain of Q1773 forms hydrogen bonds with the backbone carbonyl oxygen of N1695 and the backbone NH of A1697 of the other monomer. Hydrogen bonds are also observed between residues of two monomers including the side chains of W1802 and R1704, and the backbone carbonyl oxygen atoms of L1696 and A1697. These

residues are relatively conserved in AviM PT and CurF DH (Supplementary Fig. 1b). The observed conformation of the excised PT domain may reflect the conformation in the complete AviM megacomplex. Interestingly, the dimer interface of AviM PT is completely different from that observed in PksA PT, of which the two monomers dimerize through the C-terminal hotdog, involving the αC helix, the β7 strand following the bent long helix linker, and the linker connecting αHD2 and β10 via hydrogen bonds and salt bridges. As shown in Supplementary Fig. 9, the side chains of K1489, T1490, K1493, D1557 and N1558, the backbone carbonyl oxygen atoms of N1554, N1556 and V1559, and the backbone NH of T1490 are involved in hydrogen bonds. The R1500 side chain of one monomer packs against the E1562 side chain of the other monomer and likely makes a salt bridge.

**Substrate tunnel of AviM PT.** The catalytic dyad of AviM PT is in the similar orientation as observed in DH domains. The H949 is from the cap loop of the N-terminal hotdog and the D1104 from the helix αHD2 of the C-terminal hotdog. The Nδ of H949 forms a hydrogen bond with the backbone carbonyl oxygen of I956, suggesting that Nε may be used as a basic nitrogen to deprotonate the substrate (Fig. 4). The D1104 is 5 Å from the H949 and the D1104A mutant shows a slightly decreased production of OSA (Fig. 2a), indicating that the D1104 of AviM PT may be involved in catalysis indirectly. In contrast, the catalytic D1543 directly polarizes the catalytic H1345 by a hydrogen bond to Nε in PksA PT and mutating D1543 results in no detectable activity (Fig. 4a, b)[10,18]. The active site is positioned at the middle of a 25 Å substrate binding tunnel that is located on top of two DHD helices and closed off by the cap loop (Fig. 4c). The substrate binding tunnel starts at the junction of the N- and C-terminal hotdog and extends into the C-terminal hotdog fold. The entrance is bounded by the βN3b, the βC3, and the C-terminal end following the βC6, while the bottom is bounded by the C-terminal of the αC and the N-terminal of the βC1. The volume of the AviM PT substrate binding tunnel (761 Å³) is as same as that of CurF DH (PDB code 3KG6, 762 Å³)[27], but is significantly smaller than that of PksA PT (PDB code 5KBZ, 1196 Å³), which contains an additional hydrophobic binding region lying at the bottom of the tunnel to accommodate the hexyl group derived from the hexanoyl-CoA starter unit[18]. Compared to the tunnel of PksA PT that is closed off at the bottom by the β strands of the C-terminal hotdog, the tunnels of AviM PT and CurF DH are open at the bottoms (Fig. 4c). Interestingly, the bottom of the tunnel is differently positioned in AviM PT and CurF DH due to the alternative orientation of the αC helix. The bottom is positioned on top of the middle of the αHD2 in AviM PT. However, in CurF DH, the bottom is positioned at different side of αC helix and located on top of the C-terminal end of the αHD2. As a result, the tunnel of AviM PT is straight while the tunnel of CurF DH is bent almost 90° at the catalytic His1720. The structure of PksA PT is a cocrystal structure with substrate mimic while the structures of AviM PT and CurF DH are apo-crystal structures. Substrate binding may induce conformational changes in substrate tunnel of AviM PT and CurF DH, resulting in favorable enzyme-substrate interactions.

**QM/MM calculation of C2–C7 cyclization catalyzed by AviM PT.** The proposed mechanism for AviM PT-catalyzed cyclization involves formation of an enolate intermediate by deprotonation of C2 and subsequent aldol addition to the C7 carbonyl (Fig. 5). Electrostatic potential surface indicates the substrate tunnel is negatively charged (Supplementary Fig. 10). The polar microenvironment favors the C7 carbonyl in its electrophilic keto

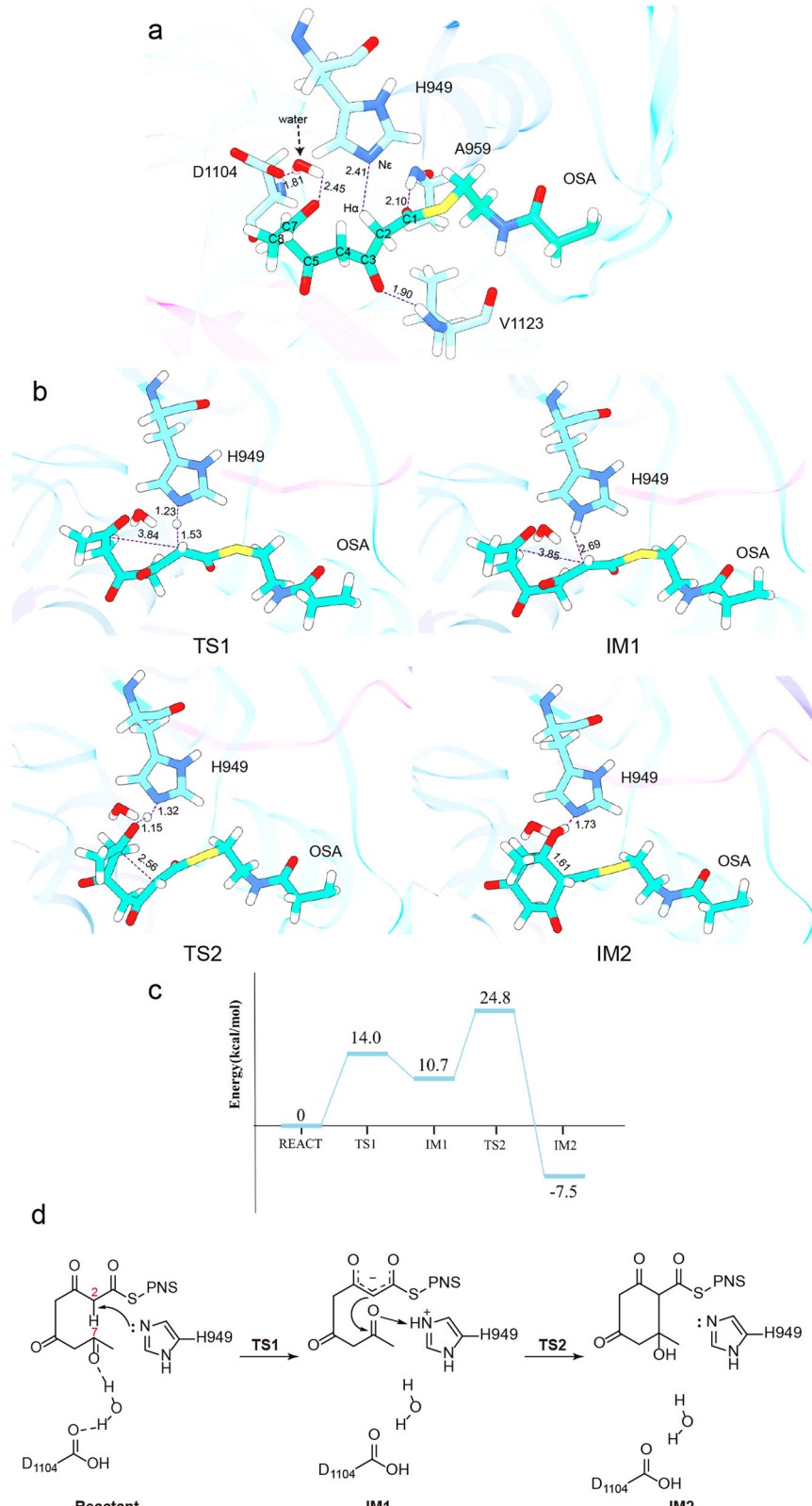

**Fig. 5 QM/MM calculation of AviM PT complexed with the substrate. a** Optimized structure of reactant. **b** Structures of TS1, IM1, TS2 and IM2. In the first stage, the C2 is deprotonated by the Nε of the catalytic H949. Next, the nucleophilic C2 launches an attack on the C7 leading to the C2–C7 cyclization. **c** The energy profiles for the deprotonation of C2 and the ring formation at the M06-2x/6-31 G(d): AMBER level. The distances are given in Å. **d** Schematic representation of the proposed mechanism for AviM PT. In reactant, the Nε of catalytic H949 abstracts proton from C2, catalytic D1104 and water molecule stabilize the substrate by hydrogen bonds. In IM1, enolate intermediate forms after deprotonation of C2. In IM2, C2–C7 aldol addiction occurs.

tautomeric form. We used QM/MM calculation to investigate the catalytic roles of His-Asp dyad and the reaction pathway of the C2–C7 cyclization (Fig. 5a). The structure of PksA PT complexed with the bisisoxazole guided the placement of the tetraketide-PPANT substrate in the active site of AviM PT[18]. The resulting complex was subjected to MD simulations. Critical reaction coordinates were analyzed to assess the important interactions between AviM PT and substrate. The distance of Nε-Hα and the angle of Nε-Hα-C2 were monitored (Supplementary Fig. 11) since the catalytic H949 utilizes its Nε as the basic nitrogen to deprotonate the substrate. The mean of d(Nε-Hα) was 3.20 Å and the average angle was 110.05°, suggesting that this system was suitable for the proton transfer. The D1104 is distant from the H949 and the substrate (>4 Å) during the MD simulations. A water molecule is positioned between the substrate C7 carbonyl and the D1104 side chain.

The "active" conformer was selected according to the distance d(Nε-Hα) and the distance d(C2–C7) and optimized at the ONIOM(M06-2X/6-31 G(d): Amber) level. In the resulting reactant, the C1 and C3 carbonyl of the substrate form hydrogen bonds with the backbone NH of A959 and V1123 respectively (Fig. 5a). The water molecule interacts with the substrate C7 carbonyl and the D1104 side chain in the reactant. As the reaction progress, the water molecule recedes from the substrate C7 carbonyl. The interaction between the water and the substrate C7 carbonyl disappears in all TS and IM structures. But the water molecule still binds to the D1104 side chain which is far away from the substrate. These results suggest that D1104 only contributes to position and stabilization of the substrate, in consistent with the fact that the D1104A mutant retains most of the activity (Fig. 1a). The D1543A mutant of PksA PT shows no detectable activity[10], indicating that D1543 may directly participates in the reaction catalyzed by PksA PT. In the first stage, the C2 is deprotonated by the H949 Nε. The first transition state (TS1) was observed when the H was located at 1.53 Å from the C2 and at 1.23 Å from the Nε (Fig. 5b). The C1–C2 and C2–C3 bonds decrease from 1.51 Å in reactant to 1.40 Å in the first intermediate (IM1), while the C = O bonds of C1 and C3 increase from 1.21 Å in reactant to 1.23 Å, suggesting the formation of an enolate structure. Consequently, the negative charge is distributed between the C2 and two adjacent carbonyls. Next, the nucleophilic C2 launches an attack on the C7. Simultaneously, the C7 carbonyl deprotonates the H949 Nε to form a hydroxy. During the process, the H of the H949 Nε gradually turns to the C7 carbonyl accompanying the decrease of the distance between C2 and C7. In TS2, the d(C2–C7) and d(H-O7) distances are 2.56 Å and 1.15 Å respectively, whereas they are further decreased to 1.56 Å and 0.99 Å in IM2. The deprotonation of C2 conquers a 14.0 kcal/mol energy barrier and the subsequent ring cyclization overcomes a 14.1 kcal/mol barrier (Fig. 5c and Supplementary Fig. 12). The two steps are feasible with low reaction barriers. Besides, the deprotonation and aldol addition are exergonic with 7.5 kcal/mol. Following aromatization could occur with a loss of hydroxide by an E1cb stepwise mechanism as proposed for PksA PT[10].

## Discussion

NR-PKSs utilize PT domains to mediate the regioselective cyclization of highly reactive poly-β-ketone intermediates. According to the phylogenetic analysis, fungal PTs are divided into 8 groups. OSAS PTs belong to the group I and perform the regioselective C2–C7 cyclization. Compare to their fungal counterparts, bacterial OSASs lack the SAT and TE domains. Our in vitro assays indicate that the AviM alone can synthesize OSA from acetyl-CoA and malonyl-CoA, but is inefficient (Fig. 2b). It's possible

that other enzymes help the initiation and/or release steps in vivo. AviM PT has low sequence similarity to fungal PTs. We confirmed the aldol cyclase of AviM PT by mutating its catalytic H949. The TTL derived from spontaneous O-C cyclization of the ACP-bound tetraketide was produced both in vivo and in vitro.

The overall structure and the active site of AviM PT resemble DHs of modular type I PKSs instead of the PksA PT. Compared to AviM PT, PksA PT uses different structural elements to form the dimer interface and has an additional helix to cap the substrate binding tunnel. The His-Asp catalytic dyad is also oriented differently. In PksA PT, the side chain of the catalytic D1543 makes a hydrogen bond with the Nε of the catalytic H1345[10]. Therefore, the polarized H1345 use its Nδ as the basic nitrogen to deprotonate the substrate. In contrast, the catalytic H949 of AviM PT is polarized by a neighboring backbone carbonyl oxygen and use its Nε as the basic nitrogen, as observed in DHs of modular type I PKSs. The corresponding D1104 is ~5 Å from the H949 and the D1104A mutant still retains most of the catalytic activity (Fig. 2a). QM/MM calculations demonstrate that the H949 initiates the reactions by abstracting a proton from C2 and transfers it to C7 carbonyl during formation of the C2–C7 bond (Fig. 5).

Iterative type I PKSs are usually found in fungi, whereas Wang et al. recently discoverers numerous iterative type I PKSs in *Streptomyces* and proposes the horizontal-gene-transfer hypothesis on the origin of fungal iterative type I PKSs[28]. To the best of our knowledge, AviM PT is the first characterized bacterial PT domain. AviM sequence was used as a seed to search homologous sequences in bacteria from Uniprot database by blast. 56 sequences of top 100 target have same domain organization as AviM (KS-AT-PT-ACP) and up to 50% sequence identity. All the sequences are from *Actinobacteria*, involving 12 genera of 6 orders. Most of them are uncharacterized. We built models of CalO5 PT and another two bacterial OSAS PT (UniProt ID: L8PIV2, A0A1V0A0I3) by using AlphaFold2[29]. The structures are similar to AviM PT in both overall structure and organization of His-Asp catalytic dyad (Supplementary Figs. 13, 14). Models of four fungal OSAS PTs were also built and showed overall similarity to PksA PT (Supplementary Fig. 15). The observation, that AviM PT structurally resembles DHs of modular type I PKSs but catalytically acts as an aldol cyclase, suggests that bacterial PTs may be evolutionary intermediates between two tailoring domains of type I PKSs. A phylogenetic tree was constructed for fungal PT, bacterial PT and DH domains. As shown in Fig. 6, fungal PTs form the first clade. A subclade of DHs of modular type I PKSs and a subclade of bacterial PTs form the second clade. The bacterial PT subclade falls between the DH subclade and the fungal PT clade.

The distinct dimer interface between bacterial and fungal PTs raised a question that whether the architecture of bacterial iPKS and fungal iPKS are different from each other. Thus, we built architectural models of AviM (KS-AT-PT-ACP) and PksA (SAT-KS-AT-PT-ACP-TE/CLC) (Supplementary Fig. 16). The model of SAT-KS-AT of PksA, and KS-AT and ACP of AviM and were built by Alphafold2. The relative position and orientation between KS-AT and the PT domains were chosen in accordance to mammalian FAS[30]. The overall architecture of AviM and PksA is similar to each other. The notable difference is that the dimeric characteristic of PT domain. In the AviM and PksA megasynthases, the distance between the C-terminal residue of AT and N-terminal residue of PT is about 25 Å and 45 Å, respectively. The longer linker (1265–1304) between AT and PT in PksA than in AviM (901–915) contributes to distant location of PksA PT (PDB code 5KBZ). Moreover, the PT-ACP linker (1661–1704) of PksA is twice that of the PT-ACP linker (1175–1196) of AviM. Additionally, PksA TE/CLC (PDB: 3ILS) is a monomer which

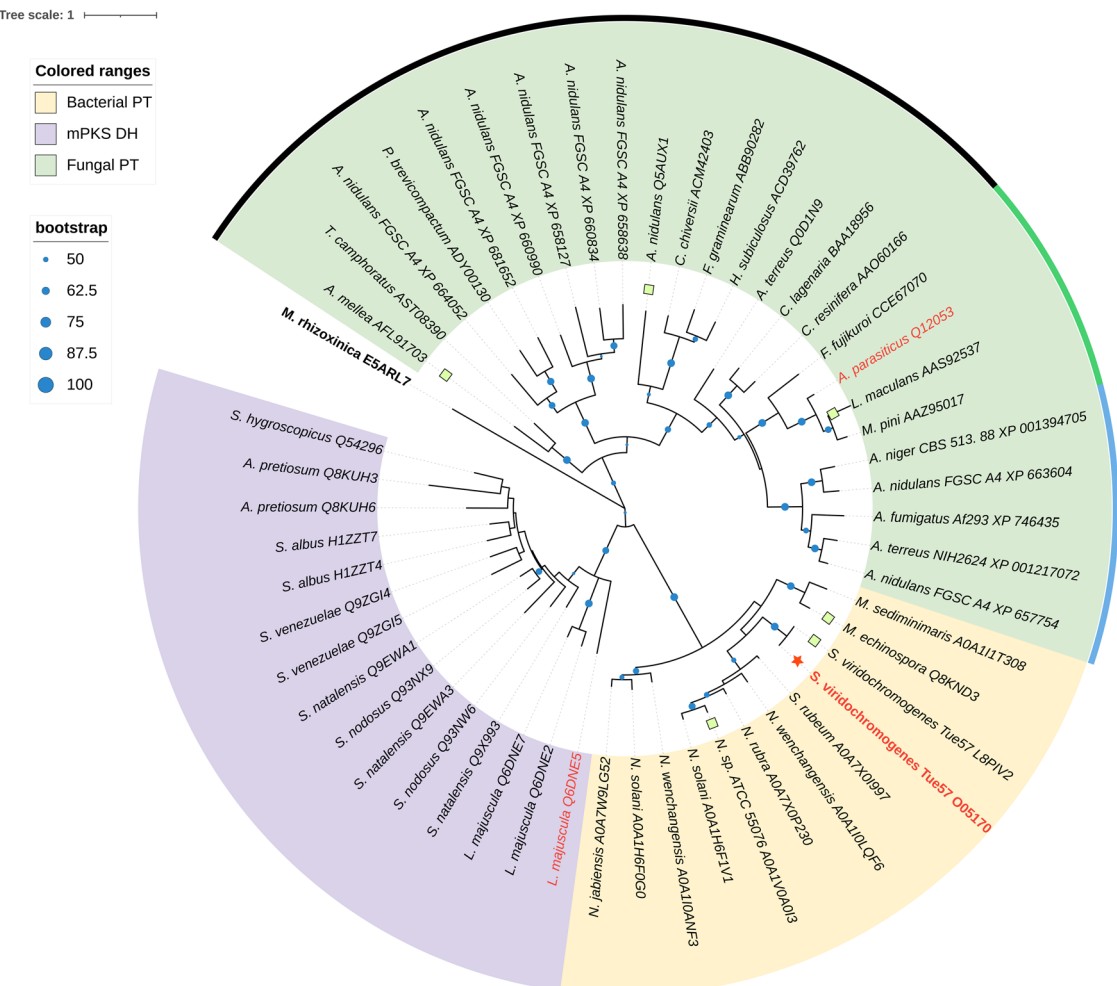

**Fig. 6 Phylogenetic tree of bacterial PTs, mPKS DHs and fungal PTs.** The color range of light yellow, light purple and pale green represent the bacterial PTs, modular type I PKS DHs and fungal PTs, respectively. The uncolored range node (*M. rhizoxinica* E5ALR7) is the outgroup branching domain. The blue dot indicates the clustering with bootstrap value. Sequence of AviM PT is marked by red asterisk and labeled in red. PksA PT (*A. parasiticus* Q12053) and CurF (*L. majuscule* Q6DNE5) are labeled in red. The black, green and dark purple circular represent fungal PTs with C2–C7, C4–C9, C6–C11 cyclization specificity.

different from the dimeric TEs from other reported NRPSs, PKSs, and mammalian FAS[31]. The longer domain-domain linkers and monomeric TE/CLC ensure that PksA PT dimerized with C-terminal structural elements other than the N-terminal β sheets.

In summary, AviM PT acts as an aldol cyclase, mediating C2–C7 cyclization in OSA biosynthesis. It's inactivation results in generation of TTL derived from spontaneous O–C cyclization of the ACP-bound tetraketide. Structural comparison reveals that the dimer interface, the overall architecture, and the catalytic dyad of AviM PT are similar to DHs of modular type I PKSs. These results suggest that bacterial PTs may represent an evolutionary intermediate between DHs and PTs of type I PKSs.

## Methods

**Materials**. Strain cultivation and DNA manipulation followed standard methods. DNA sequencing and all primers were purchased from Tsingke Biotechnology Co., Ltd. (China). Primers used for PCR and site-directed mutagenesis are listed in Supplementary Table 1. All Fast Digest enzymes were from Thermo Scientific. DNA polymerases were purchased from Invitrogen Trading Co., Ltd. (China). All the other chemicals and reagents were purchased from commercial suppliers and used without further purification. ClustalX was used in protein sequence alignments.

**Cloning and site-directed mutagenesis**. *aviM* was cloned from *S. viridochromogenes* Tü57 and digested with *Nde*I and *Hind*III restriction enzymes. The 3.9 kb DNA fragment was purified and ligated into pET28a to generate pET28-AviM. *aviM* PT was cloned and digested with *Nde*I and *Eco*RI restriction enzymes to obtain an 816 bp gene fragment, which was purified and ligated into pET28a to generate pET28-aviM-PT. The *sfp* gene was amplified from the genomic DNA of *Bacillus subtilis*. All constructs contain His$_6$ N-terminal tags for purification.

Site-directed mutagenesis was used to generate H949A, H949F, D1104A, and H949A/D1104A mutants of AviM by following standard protocol of GeneTailor™ Site-Directed Mutagenesis System (Invitrogen). A deletion mutation (AviM ΔPT) was generated by deleting PT domain from AviM via inverse PCR. The pET28-AviM was used as template and removed by *Dpn*I digestion at 37 °C for 1.5 hours. All primers with underlined mutation sites were listed in Supplementary Table 1. All plasmids were confirmed by DNA sequencing.

**Protein expression and purification**. Protein expression and purification of AviM PT domain, AviM and AviM H949A mutant follows similar protocol. The *E. coli* BL21(DE3) cells transformed with corresponding plasmids were grown to an OD$_{600}$ of 0.4 in LB medium supplemented with 50 mg/mL kanamycin at 37 °C and induced with 0.3 mM IPTG for 12 h at 16 °C. The cells were harvested by centrifugation and resuspended in lysis buffer containing 500 mM NaCl, 50 mM Tris (pH 7.5). Following sonication, cell debris was removed by centrifugation at 15,000 g for 40 minutes. The supernatant was poured over a column of Nickel-NTA resin, which was then washed with 50 mL lysis buffer containing 30 mM imidazole and eluted with lysis buffer containing 300 mM imidazole. Proteins were further purified by using a Superdex 200/Superose 6 gel filtration column (GE

Healthcare Life Sciences) equilibrated with buffer containing 150 mM NaCl and 10 mM Tris (pH 7.5). All purification steps were carried out at 4 °C.

**Heterologous production of OSA**. *E. coli* K207-3 was transformed with pET28a and its AviM derivates. *E. coli* K207-3 containing pET-28a does not lead to any OSA production and served as a negative control. The corresponding overnight pre-culture was used for inoculation of 50 mL of LB medium. 1 mM isopropyl-β-D-1-thiogalactopyranoside (IPTG) was supplemented at $OD_{600}$ of 0.4. After 24 h of incubation at 30 °C and 220 rpm, the culture was acidified with concentrated HCl to pH 4 and extracted with ethyl acetate (v/v, 1:1). The organic layer was dried and dissolved in 0.5 mL of methanol. LC–MS was conducted with an Agilent 1290 Infinity Liquid chromatography and 6545 Quadrupole Time-of-Flight Mass Spectrometer by using negative electrospray ionization and a C18 reverse-phase column (TC-C18(2) 4.6 × 250 mm, 5 μm, Agilent). The flow rate was 0.4 mL/min, and a binary solvent system gradient of 0.1% formic acid in water (A) and in acetonitrile (B) was used as follows: 0-10 min: 20–40% B, 10–30 min: 40-50% B, 30–32 min: 50–100% B, 32–35 min: 100% B, 35–36 min: 100–20% B, 36–45 min: 20% B. For MS/MS, HPLC system was coupled to a 6545XT AdvanceBio LC/Q-TOF mass spectrometer (Agilent Technologies) equipped with an ESI interface (Agilent Technologies) operating in negative ion mode using a capillary voltage of +4 kV. Other parameters were drying gas temperature, 325 °C; drying gas flow, 13 L/min; and nebulizing gas pressure, 2 bar. Detection was carried out within a mass range of 50–1100 *m/z*. The MS/MS analyses were acquired by precursor ion scan: *m/z* 167.035, RT: 12.5 min, dwell time: 1 min. CE at 2 ev, 10 ev, 20 ev and 40 ev.

**In vitro enzymatic assay of AviM**. For the enzymatic assays of AviM, a typical 100 μl reaction mixture contains 10 mM $MgCl_2$, 0.2 mM CoA, 50 μM WT AviM or its mutant and 10 μM sfp in 10 mM Tris, pH 7.4, 150 mM NaCl, and 10% glycerol. After incubation at 30 °C for 1 h, the reaction mixture was supplemented with 1 μl of acetyl-CoA (100 mM) and 2 μl of malonyl-CoA (100 mM) and incubated at 25 °C. The reaction was quenched with 5 μl of 6 mM HCl and the solution was vortexed to precipitate the enzyme. Then the mixture was spun at 14,800 rpm for 10 min and the supernatant was subjected to LC–MS analysis. The method for LC–MS is as described as in vivo analysis.

**TTL synthesis**. The synthesis of TTL has been described[32]. 2,2,6-Trimethyl-4H-1,3-dioxin-4-one **1** (3 ml, 22.5 mmol, 1.0 eq) and 1H-Benzotriazole **2** (1.0eq) in dry PhMe (80 mL) were heated for 11 h at 90 °C. Solvents were removed by rotary evaporator and a pale yellow solid **3** was generated. **1** (3.0 eq) was added dropwise to $LiN(SiMe_3)_2$ (3.1eq) in THF (750 ml) at −78 °C and stirred for 1.5 h. Then, **3** was added and the mixture was warmed up to room temperature and stirred overnight. The reaction mixture was quenched with 1 M aqueous HCl to pH 4. The mixture was extracted with EtOAc (3 × 250 ml), followed by washing with brine and drying with $MgSO_4$. The solvent was removed by rotary evaporator and the residual was dissolved in $CH_2Cl_2$ (250 ml) and washed with the Sigma-Aldrich 456101 buffer solution (pH 9.0, 5 × 50 ml). The organic layer was dried with $MgSO_4$ and the solvents were removed by rotary evaporator. The crude product was purified by silica gel column chromatography with hexanes/EtOAc (8:1 to 2:1) to give diketo-1,3-dioxinone **4** as a clear oil. **4** (100 mg, 0.44 mmol) in PhMe (30 ml) was heated for 8 h at 110 °C. The solvents were removed by rotary evaporator and the products were separated by silica gel column chromatography (hexanes:EtOAc = 2:1) to provide **TTL** as a yellowish oil. **TTL**: $^1$H NMR (400 MHz, DMSO-d6): δ (ppm) 2.17 (s, 3H), 3.75 (s, 2H), 5.27 (d, *J* = 4 Hz, 1H), 6.05(d, *J* = 4 Hz, 1H); $^{13}$C NMR (100 MHz, DMSO-d6): δ (ppm) 30.23, 47.73, 89.38, 103.08, 160.35, 164.20, 170.66, 203.31.

**Phylogenetic analysis**. A dataset of 53 sequences was selected, including 37 PTs (12 from bacteria and 25 from fungi), 15 DHs from mPKS and a branching domain from *rhizoxin* PKS as the outgroup[33]. The sequences were aligned with MAFFT-linsi (v7.480)[34] and trimmed with trimAL (v1.2)[35]. The final alignments have 278 trimmed columns. A phylogenetic tree was constructed by IQ-TREE (v1.6.12)[36] with "LG + F + G4" model and 1000 ultrafast bootstraps. The phylogenetic tree is edited by iTol.

**Crystallization and structure determination**. Crystallization of the Avim PT was performed using 1 μl of protein soulution (8 mg/mL in 150 mM sodium chloride, 10 mM Tris-HCl pH 7.5) mixed with 1 μl of reservoir solution by sitting drop vapor diffusion at 20 °C. The best crystals were obtained in a solution containing 0.1 M Tris-HCl (pH 8.5), 1 M lithium chloride, 28% PEG 4000 (w/v) within 5 days. Selenomethionyl crystals were obtained in a similar condition. The seleno-methionyl and native data were collected at wavelengths of 0.97853 Å at 100 K. All the datasets were collected at the beamline BL-18U1, Shanghai Synchrondron Radiation Facility (SSRF), China, and processed using HKL2000. Crystals belonged to space group $P2_1$ with similar unit-cell dimensions for the native and seleno-methionyl crystals. Experimental phases were determined using AutoSol in Phenix. The model was iteratively built in Coot[37] and refined in Refmac[38].

**Computational details**. The tetraketide-PPANT substrate was docked into the active site of AviM PT by AutoDock software[39]. The parameters for tetraketide-PPANT were generated at the HF/6-31 G(d) level with the Gaussian 16 program[40]. The restrained electrostatic potential (RESP)[41] charge fitting procedure was employed on the tetraketide-PPANT and the missing parameters were generated by the Antechamber package. The complex was solvated in a cubic box of TIP3P water molecules with 10 Å thickness and $Na^+$ counterions were added to achieve charge neutralization. We utilized the H + + web server to assign the protonation state of AviM PT residues at pH 7.0[42]. MD simulations of the complex were run with the AMBER software package (version 18)[43]. The enzymes and substrate were described under ff12SB and GAFF force fields from the AMBER18 software package. The Particle Mesh Ewald (PME)[44] method was conducted to account for long-range electrostatic interactions. The SHAKE algorithm was employed to fix angles and bonds involving hydrogen atoms[45]. Prior to production MD simulations, the solvated system was treated with minimization and progressive heating, with the system temperature increasing from 0 to 298 K over 50 ps. After 50 ps equilibration, the complex conducted six-times 50 ns production simulations with a different random number without any restraints under NPT conditions. Then, the conformation that favored the reaction taking place was selected as the initial structure for the QM/MM calculations. The QM calculations were performed with the M06-2×/6-31 G(d) method and the MM region was under the AMBER Parm99 force field, utilizing a two-layered ONIOM method in the Gaussian16 program. We selected the side chain of the H949, D1104 as well as a portion of the backbone of A959 and V1123. We also selected the entire tetraketide, the sulfur atom and the two adjacent CH2 groups of the substrate as QM region. Additionally, the $H_2O$ that was positioned between the substrate C7 carbonyl and the D1104 side chain was retained at the QM layer. The rest of the system was treated as the MM region.

**Statistics and reproducibility**. All the experiments in this work have been reproduced at least 2 times (*N* = 3) unless specified in the figure legends. Consistent results were obtained. Figures depicting molecular structures were constructed using UCSF Chimera[46].

**Reporting summary**. Further information on research design is available in the Nature Research Reporting Summary linked to this article.

## Data availability
Atomic coordinate has been deposited in PDB with accession codes 7VWK. The protein sequences for phylogenetic analysis were obtained from Uniprot and Genbank (Supplementary Table 2). Raw data (mass spectrum, chromatogram, NMR) is available upon request to the authors.

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

## Acknowledgements

This work was supported by National Key R&D Program of China (2020YFA0907900) and National Natural Science Foundation of China (32070040). We thank Prof. Qianjin Kang for helpful discussion about the TTL identification and the Shanghai Synchrotron Radiation Facility beamlines BL18U1 for diffraction data collection.

## Author contributions

Experimental design: J.T.Z. Experimental execution: Y.Y.F., X.Y. Computational calculation: H.N.J. Data analysis: Y.Y.F., X.Y., H.N.J., J.T.Z. Figure preparation, manuscript writing: Y.Y.F., X.Y., H.N.J., J.T.Z. Editing: Y.Y.F., X.Y., H.N.J., Z.X.D., S.J.L., J.T.Z.

## Competing interests

The authors declare no competing interests.
