## [Peer Review File · Communications Biology]

Reviewers' comments:

Reviewer #1 (Remarks to the Author):

Product template (PT) domains are used by Polyketide synthases (PKS) to catalyse the last step of the biosynthesis, which is cyclization. PT domains are mainly found in fungal PKS and PksA-PT is the best studied member. In bacteria typical, the final polyketide is released/cyclized by a TE domain. The PT domains comprise a double hot dog fold which is also shared with dehydratase domains (DH). Both enzymes share a related enzymatic mechanism. PT and DH proteins are dimeric, the main difference between both enzymes is the dimerization interface, which is basically 180° opposite. The manuscript by Feng et al describes the expression, purification of AviM, a PT domain from the bacterial *Streptomyces* species. In an enzymatic assay, the authors show the C2-C7 cyclization activity of AviM and show that His949 and Asp1104 are functionally relevant. The authors determined the crystal structure of AviM and compare it to PksA-PT and the dehydratase domain of CurF. AviM is most similar to CurF. Additionally, the authors used QM/MM simulations to analyze and propose an enzymatic mechanism. The enzymatic assay and the structure determination are sound. The QM/MM simulations and the enzyme mechanism need more careful evaluation. Figures explaining the crystal structures should be improved for better visibility.

Key Finding: That AviM-PT is more related to dehydratases than fungal PT is an interesting finding as it raises the question about the origin of PT domains and secondly the overall architecture of the polyketide megaenzymes. Do fungal PKS and modular PKS with PT domain have a different domain architecture? Maybe you want/should include a possible model for AviM (bacterial) and PksA (fungal) in the last figure? Thirdly, is the enzyme mechanism conserved in fungal and bacterial PT domains. A question that is unanswered is, if AviM is an exception or if it is likely that all bacterial PT domains share the same structure/dimerization/properties.

Main Points:

Introduction, first paragraph: please provide more references for your statements on modular type I PKS and iterative type PKS

Introduction, line 75: "PT-specific sequence insertion involved in dimerization". Please indicate this motif in your sequence alignment that the reader can check and see if AviM has this type of sequence or not.

Introduction, line 103: I am unsure if your results (simulations) are that strong to say that Ne instead of Nd is used for proton subtraction.

Introduction, line 107: "may represent an evolutionary intermediate between the two tailoring domains of type I PKS" . this sentence is not in line with Fig. S8 Your phylogenetic tree in Fig. S8 shows that AviM is a subclass of bacterial dehydratase and not an evolutionary intermediate. Fungal PTs are far more distant according to Fig. S8. Therefore, the title of the paper is wrong. If you want to make a strong statement of an "evolutionary Intermediate", then a more detailed bioinformatic analysis is necessary! The supplementary Figure S8 is insufficient.

In the introduction, you should already do a more detailed comparison of PT and DH domains. What are the similarities what are the differences? What is the catalytic mechanism of a DH domain? Which active site residues does a DH domain use? Are these residues the same as the PT domains use or are they different? To guide your reader. Because later it's important to know how DH domains operate. Maybe you want to show in the introduction already present the proposed/in the literature discussed mechanisms for PT and DH domains.

In the introduction, can you please also provide more examples of bacterial PT domains. Are there other examples known? How do those products of those PT domains look like? Is there a similarity between bacterial Pts? Do they catalyze the same mechanism or different cyclization's? This would be important for your discussion at the end, if your results are "general" or "system-specific"

QM/MM: TO me it's a bit unclear what we learn from the MM simulations? Can you please clarify and explain more? What was the goal? How do the results compare to previous experiments in the

literature?

To me, your Scheme 1 seems to be an oversimplification. Waters must be involved in substrate coordination and proton transfer reactions. I also cannot see those essential water molecules in the Figure 5?

Figure 3; The labels in the Figure are hardly visible. An additional panel showing the residues of the binding site would be helpful for understanding. To me, it seems that CurF and AviM use the same residues for Dimerization. It would also be important to label the residues that form the dimerization site in CurF, AviM and PksA-PT in the sequence alignment (Fig. S1). A superposition of the dimers would be important. Until now it's hard for the reader to understand how different or similar this is.

Figure 4: Please indicate the color used also in the legend. The tunnel is hardly visible, can you please use different colors. In general, the tunnels are very specific for the respective substrate and sometimes blocked when no substrate is present, therefore interpretations of Apo-crystal structures must be conservative.

Minor Points:

Fig S1. Please indicate the residues of PksA that build the dimerization interface.

Fig S4. Panel C and D. Can you use the same view for both proteins, it's impossible to compare both dimer interfaces.

Fig S8 The phylogenetic tree is important, maybe you want to include it in the main text as a subpanel. It shows that bacterial PTs are more related to bacterial dehydratases than to fungal PTs, in line with your structural data.

Table S2: Please indicate the phasing power for the SeMet-data or other criteria for the anomalous data quality. Where does the name Native 316-8 come from, please use a common name or the PDB identifier?

Reviewer #2 (Remarks to the Author):

The manuscript entitled "Bacterial Product Template 1 Domain: An Evolutionary Intermediate between Dehydratase and Aldol Cyclase of Type I Polyketide Synthases" by Yuanyuan Feng et al. targeted to characterize the product formation of wt and mutated AviM PT using mass spectrometry methods. Additionally, the authors performed X-ray crystallography to determine the atomic structure of the AviM PT domain, which is similar to dehydrase (DH) domains of modular type I PKSs. The authors also performed QM/MM calculations to demonstrate the mechanism of product formation.

This study is performed properly. However, some findings are not very clear. My questions are listed below:

Major comments:

1. The authors represented figure 3, the overall structure of AviM PT. and PksA PT. Did the authors calculate the structure of AviM PT and PksA PT? From figure legends, it is very difficult to understand. However, the method shows authors cloned only AviM. If authors used any other published structure, they should mention the PDB ID and write it properly in figure legends.
2. Site-directed mutations were implemented to monitor the product formation. How many new variants do they generate? Are all of them site-directed mutations, or some deletion mutations are also there? If site-directed mutations are implemented, it will be better to underline the gene sequence where the mutation is done. Rewrite the method section correctly.
3. No purification profiles (SEC) and SDS-PAGE are shown for mutated proteins. The size-exclusion

chromatography profile for wt AviM PT is very wide (Figure S5). How pure is the SEC purified protein? No SDS PAGE data showed to assess the quality of the protein.

4. It will be better if the authors represent figure 3A properly. This is the main part of the manuscript. Therefore, it will be better if the authors demonstrate the types of interactions present in the dimer interface of AviM PT?

5. Figure legends of Fig 2 should be rewritten, Or the figure should be leveled properly. It is not clear from figure legends at all. What variants do authors use for the study? What are they measuring here? The authors mentioned -- Mutating the catalytic H949 abolishes OSA production. But what about other mutations, where a small amount of OSA is produced by the mutated AviM PT?

6. In-Line number authors mentioned, "Analysis of the LC-MS data of the wild type AviM also revealed the presence of TTL,"... However, from figure2A, it is not clear the role of AviMAsp1104Ala. AviMAsp1104Ala constructs show a similar profile like wt. Similarly, what is the role of AviMHis949Ala/AviMAsp1104Ala? Basically, poor representation of the figure legends, the profile of AviMHis949Ala/AviMAsp1104Ala (yellow profile) is very confusing.

7. Figure S4 A-B should be represented in an enlarged view with two different orientations. In specially, Comparison of CurF and AviM PT, the authors highlighted that the 310 helix is differently positioned. However, some other differences are clearly visible in this figure. But authors don't mention this in the figure. Thus, authors should represent the figure at different orientations and highlight the differences.

Minor comments:

1. Line 328 "detectable activity (Figure 4x)"..... What is figure 4x?
2. In pdf file, Line 229 "E. coli B derived strain developed for the heterologous.....", some formatting issues are there.
3. The structure is similar to AviM PT both in overall structure and in organization of His-Asp catalytic dyad (Figure S7). However, from figure S7, differences & similarity is not very clear. Figure S7A and Figure S7C-F is not same scale.
4. Figures appearing in the manuscript is not numbered serially.

Reviewer #1:

Product template (PT) domains are used by Polyketide synthases (PKS) to catalyse the last step of the biosynthesis, which is cyclization. PT domains are mainly found in fungal PKS and PksA-PT is the best studied member. In bacteria typically, the final polyketide is released/cyclized by a TE domain. The PT domains comprise a double hot dog fold which is also shared with dehydratase domains (DH). Both enzymes share a related enzymatic mechanism. PT and DH proteins are dimeric, the main difference between both enzymes is the dimerization interface, which is basically 180° opposite. The manuscript by Feng et al describes the expression, purification of AviM, a PT domain from the bacterial *Streptomyces* species. In an enzymatic assay, the authors show the C2-C7 cyclization activity of AviM and show that His949 and Asp1104 are functionally relevant. The authors determined the crystal structure of AviM and compare it to PksA-PT and the dehydratase domain of CurF. AviM is most similar to CurF. Additionally, the authors used QM/MM simulations to analyze and propose an enzymatic mechanism. The enzymatic assay and the structure determination are sound. The QM/MM simulations and the enzyme mechanism need more careful evaluation. Figures explaining the crystal structures should be improved for better visibility.

Key Finding: That AviM-PT is more related to dehydratases than fungal PT is an interesting finding as it raises the question about the origin of PT domains and secondly the overall architecture of the polyketide megaenzymes. Do fungal PKS and modular PKS with PT domain have a different domain architecture? Maybe you want/should include a possible model for AviM (bacterial) and PksA (fungal) in the last figure? Thirdly, is the enzyme mechanism conserved in fungal and bacterial PT domains. A question that is unanswered is, if AviM is an exception or if it is likely that all bacterial PT domains share the same structure/dimerization/properties

Response: Thank you very much for reviewing our manuscript. We completely agree with your comments and have done our best to revise the manuscript.

The dimeric interfaces of fungal PT and bacterial PT are different suggesting that

fungal PKS and modular PKS with PT domain may have a different domain architecture. We built architectural models of AviM (KS-AT-PT-ACP) and PksA (SAT-KS-AT-PT-ACP-TE/CLC). The models of SAT-KS-AT of PksA, and KS-AT and ACP of AviM ACPs were built by Alphafold2. The relative position and orientation between KS-AT and the PT domains were chosen in accordance to mammalian FAS. The overall architecture of AviM and PksA is similar to each other. The notable difference is that the dimeric characteristic of PT domain. In the AviM and PksA megasynthases, the distance between the C-terminal residue of AT and N-terminal residue of PT is about 25 Å and 45 Å, respectively. The longer linker (1265-1304) between AT and PT in PksA than in AviM (901-915) contributes to distant location of PksA PT (PDB code 5KBZ). Moreover, the PT-ACP linker (1661-1704) of PksA is twice that of the PT-ACP linker (1175-1196) of AviM. Additionally, PksA TE/CLC (PDB: 3ILS) is a monomer which different from the dimeric TEs from other reported NRPSs, PKSs, and mammalian FAS. The longer domain-domain linkers and monomeric TE/CLC ensure that PksA PT dimerized with C-terminal structural elements other than the N-terminal β sheets (lines 450-464, Figure S16).

The overall enzyme mechanism is conserved in fungal and bacterial PT domains. But the details are different. Both of them using the catalytic His to abstract a proton. However, the function of the catalytic Asp is different. The D1104 is 5 Å from the H949 and the D1104A mutant shows a slightly decreased production of OSA (Figure 2A), indicating that the D1104 of AviM PT may be involved in catalysis indirectly. In contrast, the catalytic D1543 directly polarizes the catalytic H1345 by a hydrogen bond to N ϵ in PksA PT and mutating D1543 results in no detectable activity (Figure 4A and 4B) (lines 349-352).

AviM PT is the first characterized bacterial PT domain. AviM sequence was used as a seed to search homologous sequences in bacteria from Uniprot database by blast. 56 sequences of top 100 target have same domain organization as AviM (KS-AT-PT-ACP) and up to 50% sequence identity. All the sequences are from *Actinobacteria*, involving 12 genera of 6 orders. Most of them are uncharacterized. We built models of CalO5 PT and another two bacterial OSAS PT (UniProt ID: L8PIV2, A0A1V0A0I3) by using

AlphaFold [42]. The structures are similar to AviM PT in both overall structure and organization of His-Asp "catalytic dyad" (Figure S13, S14) (lines 436-443).

Response to each point is listed below.

Main Points:

1: Introduction, first paragraph: please provide more references for your statements on modular type I PKS and iterative type PKS

Response: We have added three references in the reference list to introduce the classification of PKSs (4. Shen, B., Polyketide biosynthesis beyond the type I, II and III polyketide synthase paradigms. *Current opinion in chemical biology*, 2003. 7(2): p. 285-295) (line 39), the organization and characteristics of modular type I PKS (3. Robbins, T., et al., Structure and mechanism of assembly line polyketide synthases. *Curr Opin Struct Biol*, 2016. 41: p. 10-18) (line 40), and the organization and characteristics of iterative PKS (5. Herbst, D.A., C.A. Townsend, and T. Maier, The architectures of iterative type I PKS and FAS. *Nat Prod Rep*, 2018. 35(10): p. 1046-1069) (line 43).

2: Introduction, line 75: "PT-specific sequence insertion involved in dimerization". Please indicate this motif in your sequence alignment that the reader can check and see if AviM has this type of sequence or not.

Response: A new Figure S1 was prepared and cited in line 75. Figure S1A shows PT-specific sequences contribute to dimerization of PT domain in fungal NR-PKSs. In Figure S1B, we labelled the interface residues of AviM PT, CurF DH and PksA PT in the sequence alignment file with orange, purple and green circle, respectively. It shows that CurF DH and AviM PT utilize similar structural elements to form dimer organization.

3: Introduction, line 103: I am unsure if your results (simulations) are that strong to say that N ϵ instead of N δ is used for proton subtraction.

Response: The PksA PT active site is a His-Asp "catalytic dyad" located at the

cyclization chamber. In a binary structure of PksA PT complexed with a bisisoxazole model of the experimentally inaccessible poly- β -ketone intermediate, the D1543 polarizes the H1345 by hydrogen bond to N ϵ so that the N δ of H1345 functions as a basic nitrogen to deprotonate the substrate in the proposed mechanism of PT-catalyzed cyclization (lines 80-84).

The proposed DH mechanism based on a crystal structure of PpsC DH (PDB code 5NJI) in complex with an α,β -double bond substrate in a *trans* configuration also involves a His-Asp “catalytic dyad”, including an α -deprotonation and a β -hydroxyl elimination stages. In the PpsC DH-catalyzed dehydration, the N δ of catalytic H959 forms a hydrogen bond with the backbone carbonyl of V996 so that the N ϵ functions as the basic nitrogen to deprotonate the substrate. The D1129 side chain acts as a general acid by interacting with the β -hydroxyl group of the substrate. ^{15}N NMR assays of *E. coli* FabA, a β -hydroxydecanoyl thioester DH, demonstrate that the catalytic H70 is held in the N δ -H tautomeric form by hydrogen bonding, whereas the N ϵ is used for proton subtraction. In a crystal structure of FabA-3-decynoyl-N-acetylcysteamine complex, the N ϵ of the catalytic H70 is covalently modified by the inactivator. QM/MM studies of mammalian FAS DH (2VZ9) also supports that the N ϵ of catalytic H887 abstract a proton from C α in a dehydration reaction line (lines 85-96).

In the AviM PT structure, the N δ of H949 is involved in a hydrogen bond with the backbone carbonyl oxygen of I956, indicating that the catalytic H949 likely uses its N ϵ as the basic nitrogen to deprotonate the substrate (lines 113-115).

4: Introduction, line 107: “may represent an evolutionary intermediate between the two tailoring domains of type I PKS”. this sentence is not in line with Fig. S8 Your phylogenetic tree in Fig. S8 shows that AviM is a subclass of bacterial dehydratase and not an evolutionary intermediate. Fungal PTs are far more distant according to Fig. S8. Therefore, the title of the paper is wrong. If you want to make a strong statement of an “evolutionary Intermediate”, then a more detailed bioinformatic analysis is necessary! The supplementary Figure S8 is insufficient.

Response: Only two bacterial PTs (AviM PT and CalO5 PT) were chosen in previous phylogenetic analysis. Now, AviM PT, CalO5 PT, and the 10 new bacterial PTs were chosen for phylogenetic analysis (Methods section, Phylogenetic Analysis, lines 204-209). As shown in Figure 6, fungal PTs form the first clade. A subclade of DHs of modular type I PKSs and a subclade of bacterial PTs form the second clade. The bacterial PT subclade falls between the DH subclade and the fungal PT clade (lines 451-454). The phylogenetic analysis, the structural similarity to DHs, and the functional similarity to PTs suggest that the bacterial PT may represent an evolutionary intermediate between DH and PT domains of type I PKSs. (lines 116-122)

5: In the introduction, you should already do a more detailed comparison of PT and DH domains. What are the similarities what are the differences? What is the catalytic mechanism of a DH domain? Which active site residues does a DH domain use? Are these residues the same as the PT domains use or are they different? To guide your reader. Because later it's important to know how DH domains operate. Maybe you want to show in the introduction already present the proposed/in the literature discussed mechanisms for PT and DH domains.

Response: As described in point 3, we compared the catalytic mechanism of PT and DH domains proposed in the literature (lines 80-96).

6: In the introduction, can you please also provide more examples of bacterial PT domains. Are there other examples known? How do those products of those PT domains look like? Is there a similarity between bacterial Pts? Do they catalyze the same mechanism or different cyclization's? This would be important for your discussion at the end, if your results are "general" or "system-specific"

Response: No bacterial PT domains has been reported yet (line 108). AviM PT is the first characterized bacterial PT domain. AviM sequence was used as a seed to search homologous sequences in bacteria from Uniprot database by blast. 56 sequences of top 100 target have same domain organization as AviM and up to 50% sequence identity. However, most of them are uncharacterized. Models of three putative PT domains

(CalO5, L8PIV2, A0A1V0A0I3) from these proteins are built by AlphaFold2. The overall structures are consistent with AviM PT. These results indicate the abundance of bacterial PTs (Discussion, lines 436-443).

7: QM/MM: TO me it's a bit unclear what we learn from the MM simulations? Can you please clarify and explain more? What was the goal? How do the results compare to previous experiments in the literature?

Response: We used QM/MM calculation to investigate the catalytic roles of "His-Asp" dyad and the reaction pathway of the C2-C7 cyclization (Figure 5) (line 377-379).

The distance of N ϵ -H α and the angle of N ϵ -H α -C2 were monitored in MD simulations since the catalytic H949 utilizes its N ϵ as the basic nitrogen to deprotonate the substrate. The mean of d(N ϵ -H α) was 3.20 Å and the average angle was 110.05°, suggesting that this system was suitable for the proton transfer (lines 382-385).

The D1104 is distant from the H949 and the substrate (>4 Å) during the MD simulations. A water molecule is positioned between the substrate C7 carbonyl and the D1104 side chain (lines 385-387). The water molecule interacts with the substrate C7 carbonyl and the D1104 side chain in the reactant optimized at the ONIOM(M06-2X/6-31G(d): Amber) level. As the reaction progress, the water molecule recedes from the substrate C7 carbonyl. The interaction between the water and the substrate C7 carbonyl disappears in all TS and IM structures. But the water molecule still tightly binds to the D1104 side chain which is far away from the substrate. These results suggest that D1104 only contributes to position and stabilization of the substrate, in consistent with the fact that the D1104A mutant retains most of the activity (Figure 1A). The D1543A mutant of PksA PT shows no detectable activity, indicating that D1543 may directly participates in the reaction catalyzed by PksA PT (lines 391-398).

The transition state and intermediate structures and energy barriers are determined (lines 398-412).

8: To me, your Scheme 1 seems to be an oversimplification. Waters must be involved in substrate coordination and proton transfer reactions. I also cannot see those essential water molecules in the Figure 5?

Response: We add the water molecular into the new scheme and Figure 5. The new scheme was incorporated into Figure 5 as panel D.

9: Figure 3; The labels in the Figure are hardly visible. An additional panel showing the residues of the binding site would be helpful for understanding. To me, it seems that CurF and AviM use the same residues for Dimerization. It would also be important to label the residues that form the dimerization site in CurF, AviM and PksA-PT in the sequence alignment (Fig. S1). A superposition of the dimers would be important. Until now it's hard for the reader to understand how different or similar this is.

Response: We adjusted the transparency of labels to make it more clearly. An additional panel C showing interface residues of AviM PT was added in Figure 3. Both AviM PT and CurF DH dimerize via β -sheets of N-terminal hotdog which was shown in Figure S7. The dimer interface residues of AviM PT, CurF DH and PksA PT are labelled in Figure S1 with orange, purple, and green circles, respectively. We superposed CurF DH and PksA PT onto AviM PT in Figure S7 and Figure S9 respectively.

10: Figure 4: Please indicate the color used also in the legend. The tunnel is hardly visible, can you please use different colors. In general, the tunnels are very specific for the respective substrate and sometimes blocked when no substrate is present, therefore interpretations of Apo-crystal structures must be conservative.

Response: We changed the substrate channel color in figure 4. The subtitle was changed to 'Substrate Tunnel of AviM PT' (line 344). We also emphasized that AviM PT was crystalized without substrate (lines 368-371). "The structure of PksA PT is a cocrystal structure with substrate mimic while the structures of AviM PT and CurF DH are apo-crystal structures. Substrate binding may induce a variety of conformational changes in substrate tunnel of AviM PT and CurF DH, resulting in favorable enzyme-substrate interactions."

Minor Points:

11: Fig S1. Please indicate the residues of PksA that build the dimerization interface

Response: We remodify the multiple sequence alignment file, the dimer interface residues of AviM PT, CurF DH and PksA PT are labelled with orange, purple and green circles, respectively.

12: Fig S4. Panel C and D. Can you use the same view for both proteins, it's impossible to compare both dimer interfaces.

Response: Figure S4 was renumbered as Figure S7. The dimeric structures of AviM PT and CurF DH were superposed.

13: Fig S8 The phylogenetic tree is important, maybe you want to include it in the main text as a subpanel. It shows that bacterial PTs are more related to bacterial dehydratases than to fungal PTs, in line with your structural data.

Response: We put the phylogenetic tree in the main text as the Figure 6.

14: Table S2: Please indicate the phasing power for the SeMet-data or other criteria for the anomalous data quality. Where does the name Native 316-8 come from, please use a common name or the PDB identifier?

Response: We add the $R_{anom}/R_{p.i.m}$ (2.00) and d''/Sig (0.66) data in Table 2 to elucidate the anomalous data quality, which calculated with SHELXD in CCP4. A common name was used for datasets.

Reviewer #2 (Remarks to the Author):

The manuscript entitled "Bacterial Product Template 1 Domain: An Evolutionary Intermediate between Dehydratase and Aldol Cyclase of Type I Polyketide Synthases" by Yuanyuan Feng et al. targeted to characterize the product formation of wt and mutated AviM PT using mass spectrometry methods. Additionally, the authors performed X-ray crystallography to determine the atomic structure of the AviM PT domain, which is similar to dehydrase (DH) domains of modular type I PKSs. The authors also performed QM/MM calculations to demonstrate the mechanism of product formation.

This study is performed properly. However, some findings are not very clear. My questions are listed below

Response: Thank you very much for reviewing our manuscript. We have done our best to revise the manuscript. Response to each point is listed below.

Major comments:

1. The authors represented figure 3, the overall structure of AviM PT. and PksA PT. Did the authors calculate the structure of AviM PT and PksA PT? From figure legends, it is very difficult to understand. However, the method shows authors cloned only AviM. If authors used any other published structure, they should mention the PDB ID and write it properly in figure legends.

Response: We solved the AviM PT structure. The PksA PT has been reported by *Barajas et al* (Proc Natl Acad Sci U S A, 2017. 114(21): p. E4142-E4148.). The PDB ID (5KBZ) is cited in the updated Figure 3 legend (line 620).

2: Site-directed mutations were implemented to monitor the product formation. How many new variants do they generate? Are all of them site-directed mutations, or some deletion mutations are also there? If site-directed mutations are implemented, it will be better to underline the gene sequence where the mutation is done. Rewrite the method section correctly.

Response: We generated three single mutations (H949F, H949A, and D1104A) and one double mutation (H949A/D1104A) of AviM by site-directed mutation. In addition, one deletion mutation (AviM Δ PT) was generated by deleting PT domain from AviM. We rewrite the method section. "Site-directed mutagenesis was used to generate H949A, H949F, D1104A, and H949A/D1104A mutants of AviM by following standard protocol of GeneTailor™ Site-Directed Mutagenesis System ((Invitrogen). A deletion mutation (AviM Δ PT) was generated by deleting PT domain from AviM via inverse PCR. The pET28-AviM was used as template and removed by DpnI digestion at 37°C for 1.5 hours. All primers with underlined mutation sites were listed in Table S1. All plasmids were confirmed by DNA sequencing." (Lines 138-143).

3: No purification profiles (SEC) and SDS-PAGE are shown for mutated proteins. The size-exclusion chromatography profile for wt AviM PT is very wide (Figure S5). How pure is the SEC purified protein? No SDS PAGE data showed to assess the quality of the protein.

Response: We prepared a new Figure S5 to show SEC and SDS-PAGE profiles of AviM and AviM H949A mutant that were purified for *in vitro* assays. Additionally, we supplemented the SDS-PAGE profile of AviM PT domain in the Figure S8.

4: It will be better if the authors represent figure 3A properly. This is the main part of the manuscript. Therefore, it will be better if the authors demonstrate the types of interactions present in the dimer interface of AviM PT?

Response: Hydrogen bonds and hydrophobic interactions contribute to dimerization of AviM PT. The figure 3 was reorganized. A new panel Figure 3C was prepared to show the interface residues of AviM PT (lines 328-330). Hydrogen bonds involved contributing to dimerization are shown as purple dashed lines. For comparison, Figure S7C was prepared to show CurF interface residues (lines 336-338) while Figure S9B was prepared to show PksA PT interface residues (lines 339-343).

5. Figure legends of Fig 2 should be rewritten, Or the figure should be leveled properly. It is not clear from figure legends at all. What variants do authors use for the study? What are they measuring here? The authors mentioned -- Mutating the catalytic H949 abolishes OSA production. But what about other mutations, where a small amount of OSA is produced by the mutated AviM PT?

Response: For *in vivo* assays, we generated three single mutations (H949F, H949A, and D1104A) and one double mutation (H949A/D1104A) of AviM by site-directed mutation. In addition, one deletion mutation (AviM Δ PT) was generated by deleting PT domain from AviM. The wild-type AviM and the H949A mutant were also assayed *in vitro*.

We rewrite the legend of Figure 2. "Function assays of AviM. (A) *In vivo* assays of AviM. *E. coli* K207-3 expressing AviM synthesizes OSA. Mutating the catalytic H949

to alanine or phenylalanine abolishes OSA production, but results in TTL from spontaneous O-C cyclization of the ACP-bound tetraketide backbone. Mutating D1104 to alanine retains most of the activity synthesizing OSA, but slightly increases TTL production. The product profile of H949A/D1104A double mutant is similar to that of H949A mutant. Deletion of PT domain from AviM abolishes OSA production whereas a small amount of TTL is still generated by the AviM Δ PT mutant. (B) *In vitro* assays of AviM. OSA is produced when AviM is incubated with acetyl-CoA and malonyl-CoA *in vitro*. TTL instead of OSA is produced by AviM H949A mutant. The traces show the extracted ion counts (m/z: 167.0350) monitored in the negative mode.”

We also updated the related second paragraph of the “Results” (lines 262-265, and 274-276).

6: In-Line number authors mentioned, "Analysis of the LC-MS data of the wild type AviM also revealed the presence of TTL,".... However, from figure2A, it is not clear the role of AviMAsp1104Ala. AviMAsp1104Ala constructs show a similar profile like wt. Similarly, what is the role of AviMHis949Ala/AviMAsp1104Ala? Basically, poor representation of the figure legends, the profile of AviMHis949Ala/AviMAsp1104Ala (yellow profile) is very confusing.

Response: We updated Figure 2 and its Figure legend. The catalytic dyad of PksA PT is composed of His1345 and Asp1543. The H949 residue of AviM PT, corresponding the catalytic H1345 of PksA PT, was replaced by site-specific mutagenesis with Ala or Phe. LC-MS analysis showed that the mutations of H949A and H949F abolished OSA production of AviM in *E. coli* K207-3, confirming the importance of H949 in catalysis (lines 253-256).

Mutating D1543 of PksA PT to alanine results in no detectable activity, whereas the corresponding D1104A mutant of AviM PT retains most of the activity. A H949A/D1104A double mutant was generated and showed product profile similar to that of H949A mutant (lines 262-265).

7. Figure S4 A-B should be represented in an enlarged view with two different

orientations. In specially, Comparison of CurF and AviM PT, the authors highlighted that the 310 helix is differently positioned. However, some other differences are clearly visible in this figure. But authors don't mention this in the figure. Thus, authors should represent the figure at different orientations and highlight the differences.

Response: A new supplementary figure (renamed as Figure S7) was prepared to show structure differences between CurF DH and AviM PT. The figure is shown at different orientation and the differences are highlighted (Figure S7A). The relative orientation of the two monomers was compared in Figure S7B. The dimer interface of CurF DH was shown in Figure S7C.

Minor comments:

1. Line 328 "detectable activity (Figure 4x)" What is figure 4x?

Response: The missing word in "detectable activity (Figure 4A and 4B)" is corrected (line 352).

2. In pdf file, Line 229 "E. coli B derived strain developed for the heterologous.....", some formatting issues are there.

Response: We amended the formatting issue (line 246).

3. The structure is similar to AviM PT both in overall structure and in organization of His-Asp ,,,"catalytic dyad"" (Figure S7). However, from figure S7, differences & similarity is not very clear. Figure S7A and Figure S7C-F is not same scale.

Response: Figure S7 renumbered as Figure S13 and be reorganized.

4. Figures appearing in the manuscript is not numbered serially.

Response: We renumber the figures in the supporting information text.

REVIEWERS' COMMENTS:

Reviewer #1 (Remarks to the Author):

I would like to thank the authors. They have sufficiently answered all the questions raised.

Reviewer #2 (Remarks to the Author):

The authors have modified the introduction and method portion appropriately. Authors modified several figures and represented the figures more appropriately, which will be easier for the general reader. Additionally, the authors explained most of the reviewer's suggestions. The authors addressed most of the questions, and their responses are convincing. However, some minor comments are given below:

Minor comments:

1. Line 144 "DpnI digestion at 143 37°C for 1.5 hours". Somehow "degree" symbol appeared as a small box?
2. No figure number (1-6) is there in the merged pdf. Very difficult to follow it.
3. In figure 3C (I guess because figure number is not there), the authors represented various H-bonding and hydrophobic interactions. However, the authors did not discuss how they calculate the bond distances or create the interacting amino acid residues.
4. In the main text in line 330: "The dimer interface of AviM PT was dominated by hydrophobic interactions and hydrogen bonds (Figure 3C)." If authors calculated the interacting amino acids, and if authors are confident, they should highlight which amino acids are involved in hydrophobic interactions and hydrogen bonds.
5. Again line 338, "Consistent with AviM PT, β sheets and β -hairpins of N337 terminal hotdog of CurF DH participated in dimerization via hydrophobic interactions and hydrogen bonds (Figure S7C)." Are these amino acids residues conserved in AviM PT and CurF DH?
6. Authors should mention which amino acids are responsible for hydrogen bonds and salt bridges in the text (Line 342 Figure S9).

Reviewer:

The authors have modified the introduction and method portion appropriately. Authors modified several figures and represented the figures more appropriately, which will be easier for the general reader. Additionally, the authors explained most of the reviewer's suggestions. The authors addressed most of the questions, and their responses are convincing. However, some minor comments are given below:

Response: Thank you very much for reviewing our manuscript. We agree with all your comments and have tried our best to revise the manuscript. The response to each point is listed as below.

Minor comments:

1. Line 144 "DpnI digestion at 37°C for 1.5 hours". Somehow "degree" symbol appeared as a small box?

Response: Fixed (line 389).

2. No figure number (1-6) is there in the merged pdf. Very difficult to follow it.

Response: Figures were provided as separated files.

3. In figure 3C (I guess because figure number is not there), the authors represented various H-bonding and hydrophobic interactions. However, the authors did not discuss how they calculate the bond distances or create the interacting amino acid residues.

Response: Residues from the two monomers AviM PT that are within 4.5 Å from each other at the binding interface were recognized as interfacial residues (lines 212-213). The distance between the donor and acceptor atoms of a hydrogen bond was defined as less than 3.5 Å (lines 216-217).

4. In the main text in line 330: "The dimer interface of AviM PT was dominated by hydrophobic interactions and hydrogen bonds (Figure 3C)." If authors calculated the interacting amino acids, and if authors are confident, they should highlight which amino acids are involved in hydrophobic interactions and hydrogen bonds.

Response: The residues of V926, L931, L933 and V999 create a hydrophobic core between the β N1, β N2 and β N4 sheets of the two monomers (lines 215-216). The distance between the donor and acceptor atoms of a hydrogen bond was defined as less than 3.5 Å. According to this criterion, several hydrogen bonds are observed between the two monomers. The side chain of Q997 forms hydrogen bonds with the backbone NH of A927, and the backbone carbonyl oxygen of G925. The backbone carbonyl oxygen of G925 also makes a hydrogen bond with the side chain of W1022, whereas the backbone carbonyl oxygen of A927 forms hydrogen bonds with the side chain of R1006 (lines 216-221).

5. Again line 338, "Consistent with AviM PT, β sheets and β -hairpins of N337 terminal hotdog of CurF DH participated in dimerization via hydrophobic interactions and hydrogen bonds (Figure S7C)." Are these amino acids residues conserved in AviM PT and CurF DH?

Response: In CurF DH, the residues I1694, L1696, I1699 of both monomers forms a hydrophobic core at the dimer interface. The side chain of Q1773 forms hydrogen bonds with the backbone carbonyl oxygen of N1695 and the backbone NH of A1697 of the other monomer. Hydrogen bonds are also observed between residues of two monomers including the side chains of W1802 and R1704, and the backbone carbonyl oxygen atoms of L1696 and A1697. These residues are relatively conserved in AviM PT and CurF DH (Supplementary Figure 1B) (lines 226-234).

6. Authors should mention which amino acids are responsible for hydrogen bonds and salt bridges in the text (Line 342 Figure S9).

Response: The residues involved in hydrogen bonds and salt bridges were described. "As shown in Supplementary Figure 9, the side chains of K1489, T1490, K1493, D1557 and N1558, the backbone carbonyl oxygen atoms of N1554, N1556 and V1559, and the backbone NH of T1490 are involved in hydrogen bonds. The R1500 side chain of one monomer packs against the E1562 side chain of the other monomer and likely makes a salt bridge (lines 238-242)."